

# Modelling the effect of curves on distance running performance

Paolo Taboga[1,*] and Rodger Kram[2,*]

[1] Kinesiology Department, California State University, Sacramento, CA, United States of America
[2] Integrative Physiology Department, University of Colorado, Boulder, CO, United States of America
[*] These authors contributed equally to this work.

## ABSTRACT

**Background**. Although straight ahead running appears to be faster, distance running races are predominately contested on tracks or roads that involve curves. How much faster could world records be run on straight courses?

**Methods**. Here, we propose a model to explain the slower times observed for races involving curves compared to straight running. For a given running velocity, on a curve, the average axial leg force ($\overline{F}_a$) of a runner is increased due to the need to exert centripetal force. The increased $\overline{F}_a$ presumably requires a greater rate of metabolic energy expenditure than straight running at the same velocity. We assumed that distance runners maintain a constant metabolic rate and thus slow down on curves accordingly. We combined published equations to estimate the change in the rate of gross metabolic energy expenditure as a function of $\overline{F}_a$, where $\overline{F}_a$ depends on curve radius and velocity, with an equation for the gross rate of oxygen uptake as a function of velocity. We compared performances between straight courses and courses with different curve radii and geometries.

**Results**. The differences between our model predictions and the actual indoor world records, are between 0.45% in 3,000 m and 1.78% in the 1,500 m for males, and 0.59% in the 5,000 m and 1.76% in the 3,000 m for females. We estimate that a 2:01:39 marathon on a 400 m track, corresponds to 2:01:32 on a straight path and to 2:02:00 on a 200 m track.

**Conclusion**. Our model predicts that compared to straight racecourses, the increased time due to curves, is notable for smaller curve radii and for faster velocities. But, for larger radii and slower speeds, the time increase is negligible and the general perception of the magnitude of the effects of curves on road racing performance is not supported by our calculations.

Corresponding author
Paolo Taboga, paolo.taboga@csus.edu

## INTRODUCTION

Although straight ahead running appears to be faster, distance running races are predominately contested on tracks or roads that involve curves. How much faster could world records be run on straight courses? Could the 2-hour marathon barrier be broken without pacers or drafting if the course was perfectly straight? Could women break the 4-minute mile barrier?

Previous analysis of curve running has focused on sprinting. Coherent explanations for slower human sprint performances on curves, based on physics and biomechanics, are supported by substantial empirical evidence. The requirement to exert centripetal force (*Greene, 1985*), and more specifically the force generated by the inside leg (*Chang & Kram, 2007*; *Churchill et al., 2016*), adequate friction/traction (*Alexander, 2002*; *Luo & Stefanyshyn, 2011*; *Luo & Stefanyshyn, 2012*), and ankle inversion/eversion torques (*Greene, 1987*; *Luo & Stefanyshyn, 2012*) have all been implicated as explanations for reduced sprint velocity along curves. Quadrupedal (greyhound) sprint performance on curves is less explored and more controversial (*Usherwood & Wilson, 2005*; *Hayati et al., 2017*). A sizable body of scientific research articles also exist regarding the mechanics and energetics of small radius turns executed by humans and other animals (e.g., *Wilson et al., 2013*) but such turns do not involve substantial centripetal forces.

Here, we focus on how curves affect the middle- and long-distance running performance of human athletes. Performances on standard indoor tracks (200 m/lap with curve radii of 17.2 m (*IAAF, 2008a*)) are generally slower than on tracks with larger radii. For example, the National Collegiate Athletics Association (NCAA) equates a 4:03.07 mile on a standard 200 m indoor track to a 4:00.00 mile on an ''oversized'' track (i.e., >300 m/lap) (*Pederson et al., 2012*) (NCAA, 2012). What are the physiological/biomechanical mechanisms responsible for this effect?

A fundamental physiological limit to distance running performance is the ability to generate adequate energy (i.e., ATP) from aerobic metabolism. Three physiological factors determine distance running performance: maximal aerobic capacity ($\dot{V}O_2$ max), the submaximal rate of oxygen uptake required to run at a specified velocity (aka ''running economy'') and the % or fraction of the maximal aerobic capacity that can be sustained (*Bassett Jr & Howley, 2000*; *Coyle, 1995*; *Joyner & Coyle, 2008*; *Ferretti, Bringard & Perini, 2011*). For example, an elite male marathon runner might have a $\dot{V}O_2$ max of 86 mlO$_2$/kg/min and the physiology to sustain 85% of that $\dot{V}O_2$ max (71ml O$_2$/kg/min) for more than 2 h at a velocity of 5.55 m/s in a straight line (*Joyner, 1991*).

The rates of oxygen uptake or metabolic energy required to run straight-ahead at a specified velocity are proportional to the force applied to the ground (*Arellano & Kram, 2014*; *Kipp, Grabowski & Kram, 2018*; *Kram & Taylor, 1990*). During distance running, the vertical ground reaction force (GRF) vs. time pattern resembles a half-sine wave with a peak magnitude of 2.5 to 4 x body weight depending on velocity. During straight-line running, the vertical GRF averaged over a complete stride is equal to 1.0 x body weight (BW). But, when a person runs along a curved path on a flat surface, they lean in towards the center of the curve and the required average force axial to the leg is greater than 1.0 BW due to the need to exert a centripetal force. The greater average axial leg force presumably requires a greater rate of metabolic energy expenditure than straight running. Centripetal force is equal to $mv^2/r$, where m is body mass, v is tangential velocity and r is the curve radius. *Hamill, Murphy & Sussman (1987)* have measured GRF during distance running on curves equivalent to an outdoor 400 m track and for $v = 6.3$ m/s (corresponding to a mile time of 4:15.00) the peak centripetal GRF is ∼0.6 BW. Accordingly, an athlete must run slower on a curved path to maintain the same metabolic energy expenditure.

Overall, our objective was to combine physics and physiology to model the energetics of running on curved paths. From these energetic cost estimates, we then calculated race time differentials for various race distances and velocities. A few distance running races occur along straight paths (e.g., the Fifth Avenue Mile in New York City) but most involve at least one turn. We begin our analysis by modelling an out-and-back racecourse with a single 180° turn. We then move to the much more common track races which are contested on standard 200 m indoor oval tracks (inner edge radius = 17.2 m) and standard 400 m outdoor oval tracks (inner edge radius = 36.5 m). According to the rules of the governing body for athletics, the IAAF (*IAAF, 2008a*; *IAAF, 2008b*), both indoor and outdoor tracks must be measured 0.3 m from the raised curb positioned at the inner edge of lane 1. The added 0.3 m takes in account the theoretical line of running (*IAAF, 2008b*) of athletes who will run on curves with actual radii of 17.5 m (*IAAF, 2008a*) and 36.8 m (*IAAF, 2008b*) on indoor and outdoor tracks respectively. We then consider races longer than 10,000 m which are predominately conducted on road surfaces. Road-racing courses typically involve multiple turns of different radii and we demonstrate how we can apply our model to any course configuration. Finally, urban races often involve negotiating city blocks comprising 90 degree angles. According to the IAAF, road racing courses must be measured 0.3 m from the curb (IAAF) which equates to 0.3 m radius. In the appendix, we consider the special case of running races on rectangular city blocks.

## METHODS

### Gross metabolic energy expenditure as a function of body weight

Using a spring and harness system, *Teunissen, Grabowski & Kram (2007)* quantified how simulated reduced gravity decreased the gross rate of metabolic energy expenditure during treadmill running. We utilized their data to calculate the fractional change in the rate of gross metabolic energy expenditure $f$ as a function of the average axial leg force ($\overline{F}_a$)

$$f = 0.6234\overline{F}_a + 0.3766 \tag{1}$$

where $\overline{F}_a$ is expressed as multiples of body weight (BW) and is calculated over an entire stride cycle (from touch down of one leg to the next touch down of the same leg). While *Teunissen, Grabowski & Kram (2007)* only measured metabolic energy expenditure in normal and simulated reduced gravity ($\overline{F}_a \leq 1BW$), we assume that the slope in Eq. (1) extrapolates to $\overline{F}_a > 1BW$. According to Eq. (1), when a person is running in a straight line ($F_a$=1 BW), $f = 1$ (i.e., no change). If $\overline{F}_a = 1.25$ (a 25% increase in average axial force), $f = 1.16$ (a 16% increase in gross metabolic energy expenditure). Note that we calculated Eq. (1) based on the data of *Teunissen, Grabowski & Kram (2007)* but to calculate the gross rate of metabolic energy expenditure, we added Teunissen et al.'s standing metabolic rate value of 1.87 W/kg to the net values reported in their tables. Further, we forced the regression to have an exact value of $f$ =1 when $\overline{F}_a$=1.

### Axial leg force as a function of velocity and curve radius

A person with a body mass $m$ (kg), running with a tangential velocity $v$ (m/s) on a curve of radius $r$ (m) is subject to two forces in the frontal plane (*Greene, 1985*): $\overline{F}_v$ the average

force in the vertical direction due to gravity,

$$\overline{F}_v = mg \qquad (2)$$

where $g$ is gravitational acceleration, and $\overline{F}_c$ the average centripetal force,

$$\overline{F}_c = \frac{mv^2}{r} \qquad (3)$$

The vector sum of $F_v$ and $F_c$ is the average axial leg force:

$$\overline{F}_a = \sqrt{\overline{F}_v^2 + \overline{F}_c^2} \qquad (4)$$

where $\overline{F}_a$ is measured in newtons (N). Dividing Eq. (4) by the body weight of the runner (1 $BW = mg$) and combining it with Eqs. (2) and (3), the average axial force $\overline{F}_a$ can be calculated in multiples of body weight:

$$\overline{F}_a = \sqrt{1 + \frac{v^4}{(gr)^2}} \qquad (5)$$

## Gross rate of metabolic energy expenditure during curve running

By inserting Eq. (5) into Eq. (1), it is possible to calculate the fractional increase, $f$, in the gross rate of metabolic energy expenditure for a runner with a tangential velocity v, on a curve of radius r compared to running straight-ahead at the same velocity:

$$f = 0.6234 \sqrt{1 + \frac{v^4}{(gr)^2}} + 0.3766 \qquad (6)$$

As $r \to \infty$ (straight running), $\overline{F}_a \to 1$ in Eq. (5) (Fig. 1A) and therefore Eq. (6) reduces to $f \to 1$ (Fig. 1C) irrespective of running velocity $v$. At slower velocities, as $v \to 0$, $\overline{F}_a \to 1$ in Eq. (5) (Fig. 1B) and therefore Eq. (6) reduces to $f \to 1$ (Fig. 1D) irrespective of curve radius $r$.

## Running velocity on straight and curved paths

The following equation, derived by *Kipp, Kram & Hoogkamer (2019)*, expresses the relationship between gross rate of oxygen uptake ($\dot{V}O_{2s}$) and overground running velocity ($v_s$) on a straight path:

$$\dot{V}O_{2s} = 0.02724v_s^3 + 1.7321v_s^2 - 0.4538v_s + 18.91 \qquad (7)$$

where $\dot{V}O_{2s}$ is measured in $mlO_2/min/kg$. The cubic term in Eq. (7) takes into account air resistance (*Pugh, 1970*). The *Kipp, Kram & Hoogkamer (2019)* equation is based on submaximal (below lactate threshold, LT), steady state measurements of oxidative metabolic rates. Beyond the lactate threshold, indirect calorimetry calculations (derived from oxygen uptake and carbon dioxide production rates) do not represent all of the metabolic energy required. However, we believe that extrapolating to faster speeds (beyond LT) provides a reasonable measure of the total metabolic energy required even though in reality that energy is supplied by both oxidative and non-oxidative metabolism.

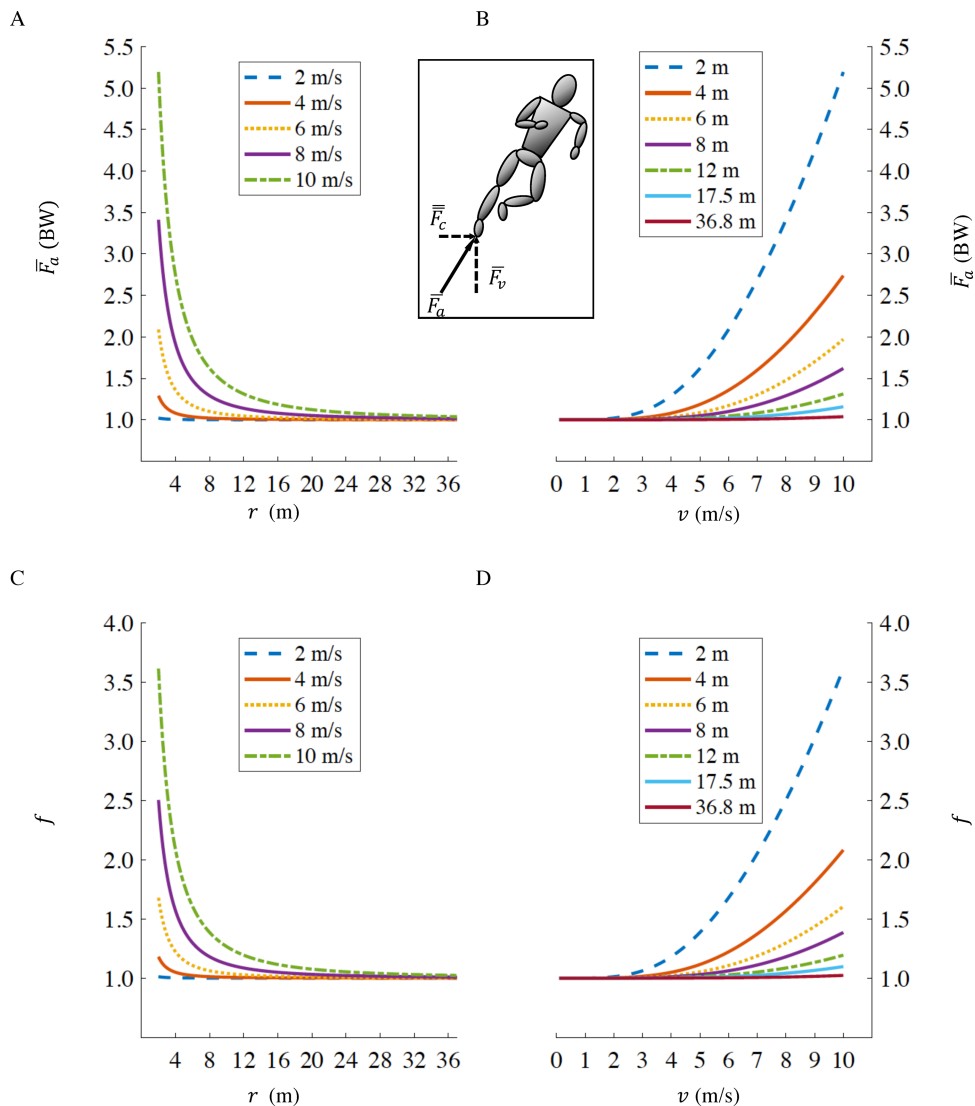

**Figure 1** Axial leg force $\overline{F}_a$ as a function of curve radius r (A), and running velocity v (B); fractional increase in gross rate of metabolic energy expenditure $f$ as a function of curve radius r (C), and velocity v (D). The standard radii for indoor and outdoor running tracks, are $r = 17.5$ m and $r = 36.8$ m respectively. Inset figure representing centripetal ($\overline{F}_c$), vertical ($\overline{F}_v$) and axial ($\overline{F}_a$) forces. Values of $\overline{F}_a$ and $f$ when running at fast velocities ($>8$ m/s) on small radii ($<6$ m) are not physiological, we depict them only to illustrate the effects of velocity and radius in extreme conditions.

To calculate the gross rate of oxygen uptake of a person running on a curve ($\dot{V}O_{2c}$) with a tangential velocity along the curve $v_c$, we can combine Eqs. (6) and (7):

$$\dot{V}O_{2c} = (0.6234\sqrt{1 + \frac{v_c^4}{(gr)^2}} + 0.3766)(0.02724v_c^3 + 1.7321v_c^2 - 0.4538v_c + 18.911) \quad (8)$$

where $\dot{V}O_{2c}$ is measured in mlO$_2$/min/kg. Note: we and others prefer to express running economy in units of energy or power (e.g., W/kg or kcals/min/kg) (*Beck et al., 2018*;

*Fletcher, Esau & Macintosh, 2009*; *Kipp, Byrnes & Kram, 2018*; *Shaw, Ingham & Folland, 2014*) to account for differences in substrate utilization and therefore, in the amount of energy liberated per liter of oxygen uptake. However, *Pugh (1970)* used oxygen uptake rates. For our purpose here, assuming equivalence between rates of metabolic energy utilization and oxygen uptake incurs an insignificant error because we are only considering small changes in metabolic rate between curve and straight-line running.

A runner maintaining a constant velocity on both straight and curved portions ($v_s = v_c$), would therefore alternate their gross rates of metabolic energy expenditure according to Eqs. (7) and (8) respectively, where $\dot{V}O_{2c} > \dot{V}O_{2s}$. A runner performing at the maximal sustainable percentage of their aerobic capacity on the straight portion of a race cannot sustain an equal tangential velocity on the curve, since this would increase their rate of metabolic energy expenditure. Rather, in order to maintain the same metabolic energy expenditure throughout the race, running velocity on the curve must be reduced ($v_c < v_s$) so that $\dot{V}O_{2c} = \dot{V}O_{2s}$.

To calculate the running velocity on the curve ($v_c$) for a given velocity on the straight ($v_s$), we used numerical approximation methods (see Appendix for algorithm 1). To calculate the increased time during a single gradual 180° turn in an out-and-back race, we first used Eq. (7) to calculate the required $\dot{V}O_{2s}$ for a straight racecourse. We then calculated the running velocity $v_c$ on the curved portion according to Eq. (8) given the same metabolic energy expenditure ($\dot{V}O_{2c} = \dot{V}O_{2s}$) for a range of radii from 0.3 m (minimum radius according to IAAF rules (IAAF)) up to 36.8 m (outdoor track (*IAAF, 2008b*)). The increased time during the curved portion is calculated as:

$$\Delta t_{180°} = \frac{d_c}{v_c} - \frac{d_c}{v_s} \tag{9}$$

where $d_c$ is the distance run in the 180° turn, corresponding to $\pi r$.

In order to calculate the time difference between a straight race course ($t_{straight}$) and the same racing distance on indoor or outdoor tracks, we used the same approach described above, i.e.: we assumed that an athlete maintains the same metabolic energy expenditure on the straight and on the curved portion of the track ($\dot{V}O_{2c} = \dot{V}O_{2s}$), with the curve radii set at 17.5 m for indoor track (*IAAF, 2008a*) and 36.8 m for the outdoor track (*IAAF, 2008b*). The total time ($t_{track}$) on the track is then calculated as:

$$t_{track} = \frac{d_s}{v_s} + \frac{d_c}{v_c} \tag{10}$$

where $d_s$ and $d_c$ are the total distances run on the straight and curved portions of the track, respectively, and the total racing distance is $d_{tot} = d_s + d_c$.

Vice-versa, when a certain time $t_{track}$ on the track is known, assuming that $\dot{V}O_{2c} = \dot{V}O_{2s}$, it is possible to calculate the respective velocities on the straight and curved paths, $v_s$ and $v_c$, that satisfy Eq. (10) (see Appendix for algorithm 2). The respective time on a straight racecourse would then be:

$$t_{straight} = \frac{d_{tot}}{v_s} \tag{11}$$

The same procedure can be used to convert times between tracks with different curve radii and/or sizes: for example between indoor ($r_{indoor} = 17.5$ m, distance of one lap $d_{lap,indoor}$=200 m) vs. outdoor tracks ($r_{outdoor} = 36.8$ m, $d_{lap,outdoor}$=400 m).

For a given racing distance, it is possible to calculate the time difference $\Delta t$ as follows:

$$\Delta t = t_{indoor} - t_{outdoor} \tag{12}$$

to compare indoor vs. outdoor tracks,
and:

$$\Delta t = t_{straight} - t_{outdoor} \tag{13}$$

to compare straight racecourses vs. outdoor tracks.

Given that $r_{indoor} < r_{outdoor}$, $\Delta t > 0$ in Eq. (12) represents the increased amount of time for running on an indoor track while keeping the same rate of oxygen uptake maintained on the outdoor track. On the other hand, given that $r_{straight} \to \infty$, $\Delta t < 0$ in Eq. (13) represents the increased amount of time for running on a straight racecourse while keeping the same rate of oxygen uptake maintained on the outdoor track. We used the outdoor 400 m track as a reference because the majority of racing distances (1,500 m, 3,000 m, 5,000 m and 10,000 m) are commonly run on outdoor tracks, compared to indoor tracks (1,500 m, 3,000 m and 5,000 m) and very few races are contested on straight racecourses.

We also determined the ideal geometry of an outdoor track, where we kept the track lap distance constant ($d_{lap,outdoor}$=400 m) and changed curve radii from $r$=6 m, corresponding to an oval track with a total straight portion of 362.3 m and a total curved portion of 37.7 m per lap, to $r = 63.66$ m, corresponding to a perfectly circular track with all 400 m run on the curved portion.

We selected the world record times $t_{WR}$ on a standard outdoor track for 1,500 m, 3,000 m, 5,000 m and 10,000 m as a reference and then calculated the total racing time $t(r)$ as a function of the different curve radii $r$ according to Eq. (10). The time difference $\Delta t$:

$$\Delta t = t(r) - t_{WR} \tag{14}$$

is the increased time ($\Delta t > 0$) or decreased time ($\Delta t < 0$) as a function of radius $r$ compared to the respective world record. The ideal track geometry corresponds to the curve radius that allows the biggest time savings.

More generally, these algorithms can be used to convert times between straight racecourses and the same distance run on a path with a series of curves with different radii:

$$t_{path} = \frac{d_s}{v_s} + \sum \frac{d_{c,i}}{v_{c,i}} \tag{15}$$

where $d_{c,i}$ is the distance ran on the i-th curve, with a given radius $r_i$, and $v_{c,i}$ is the velocity on the i-th curve.

## Breaking 2 on a straight path

Using Eq. (15), we analyzed the racetrack in Monza, Italy used for the "Breaking 2" marathon exhibition (https://en.wikipedia.org/wiki/Breaking2). We divided the total

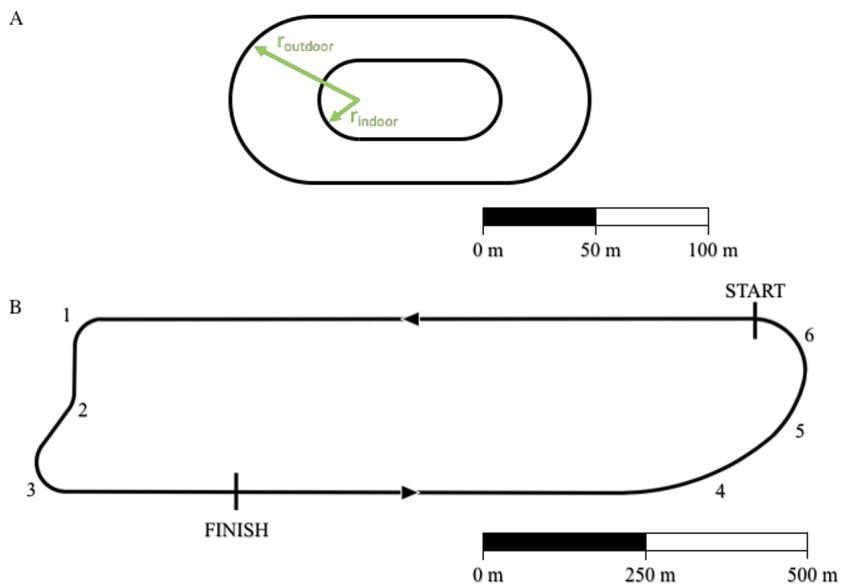

**Figure 2  Outline of standard outdoor and indoor tracks (A) and of the Monza racetrack utilized during the "Breaking 2" project (B).** For indoor track: $r_{indoor} = 17.5$ m and the distance of one lap $d_{lap,indoor} = 200$ m. For outdoor track: $r_{outdoor} = 36.8$ m and $d_{lap,outdoor} = 400$ m. We divided the south curve ("Curva parabolica") into three different portions (labeled 4, 5 and 6) in order to account for the non-constant radius of this specific section. A and B have different scales.

lap distance, $d_{lap,Monza} = 2424.4$ m (17.4 laps to run a full marathon: 42,195 m) into a straight portion $d_s$=1907 m and 6 different curves (see Fig. 2, racetrack blueprints: personal communication, Brett Kirby, Ph.D.). We divided the "Curva parabolica" into three different portions in order to account for the non-constant radius of this specific section. All other curves were assumed to have a fixed radius throughout each section. We then applied the same algorithm described in the previous paragraph: we calculated the running velocities on the straight and on each of the curved portions of the track assuming that Eliud Kipchoge maintained a constant $\dot{V}O_{2s} = \dot{V}O_{2c}$. We then converted the total time $t_{Monza}$ =7225 s (2 h and 25 s) to the time $t_{straight}$ that Kipchoge might have run on a straight path with a length $d_{tot}$=42195 m, while maintaining all the other factors (drafting, shoes, hydration etc.) adopted during the Breaking 2 attempt.

## RESULTS

### Increased time for a single 180° turn

We report the increased time $\Delta t_{180°}$ as a function of radius $r$ according to Eq. (9) for three different representative velocities ($v_1 = 7.3$ m/s, corresponding to Hicham el Guerrouj's 1,500 m world record; $v_2$ =6 m/s, corresponding to the men's half marathon world record; and a recreational running velocity, $v_3$ =4 m/s) in Fig. 3. The radius $r$ influences both the distance run on the curve $d_c$ and the velocity on the curve $v_c$ in Eq. (9). As $r \to \infty$, $d_c \to \infty$, but given that $v_c \to v_s$ (Eq. (8)), the increased time $\Delta t_{180°} \to 0$. As $r$ decreases, $v_c < v_s$ and $\Delta t_{180°}$ starts to increase up to a specific radius $\hat{r}$, different for each velocity. In

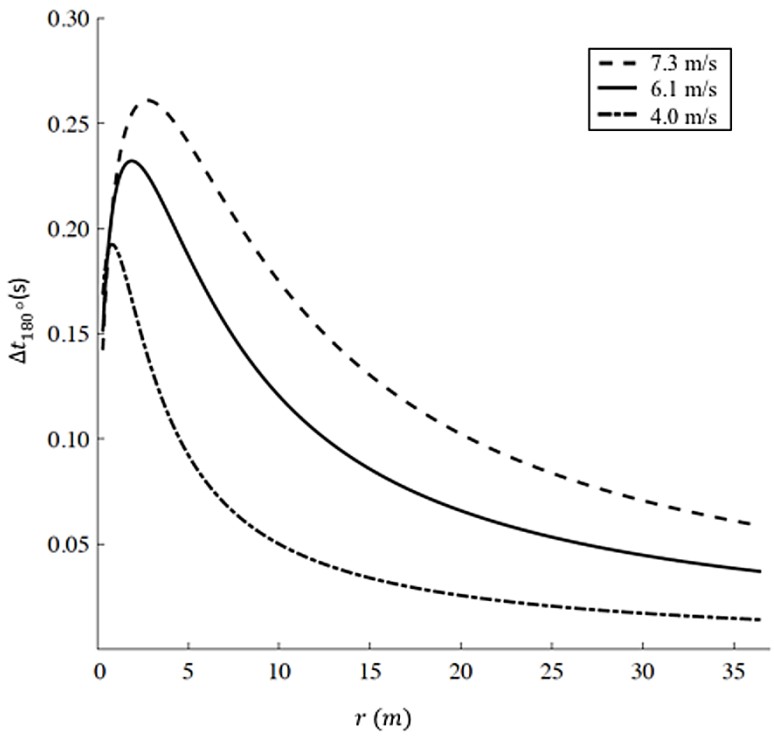

**Figure 3 Increased time for a 180° turn as a function of radius.** We selected three velocities: $v_1 = 7.3$ m/s (dashed line, corresponding to Hicham el Guerrouj's 1,500 m world record), $v_2 = 6.1$ m/s (continuous line, corresponding to men's half marathon world record) and $v_3 = 4$ m/s (dash-dotted line). For each line, the maximum Delta t is located at $r_1 = 2.8$ m, $r_2 = 1.9$ m and $r_3 = 0.8$ m for $v_1$, $v_2$ and $v_3$ respectively. Caution should be used when applying our model at very small radii (<6 m). *Chang & Kram (2007)* report a maximum *sprinting* velocity $v = 5.66$ m/s for $r = 6$ m, while our model predicts an unrealistic sustained velocity $v_c = 6.71$ m/s.

particular $\hat{r}_1 = 2.7$ m and $\Delta t_{180°} = 0.261$ s for $v_1$; $\hat{r}_2 = 1.9$ m and $\Delta t_{180°} = 0.232$ s for $v_2$; $\hat{r}_3 = 0.8$ m and $\Delta t_{180°} = 0.193$ s for $v_3$. $\hat{r}$, therefore, represents the worst radius in terms of velocity reduction ($v_c < v_s$) and non-trivial distance run on the curve ($d_c > 0$). As $r$ further decreases ($r < \hat{r}$), $d_c \to 0$, leading to an overall decrease in $\Delta t_{180°}$.

### Outdoor tracks vs. indoor tracks vs. straight races

We report the time difference $\Delta t$ a function of running velocity $v$ in Fig. 4A (1,500 m, 5,000 m and 10,000 m) and Fig. 4B (half marathon and marathon) respectively, the maximum velocity $v$ for each distance corresponds to the respective current men's world record. In both figures, $\Delta t > 0$ represents the increased amount of time for running on an indoor track ($r = 17.5$ m) compared to the outdoor track ($r = 36.8$ m), while $\Delta t < 0$ represents the decreased amount of time for running on a straight racecourse compared to an outdoor track. The increased or decreased amount of time compared to an outdoor track increases non-linearly with velocity $v$ and is inversely proportional to the curve radius $r$ (see Appendix for step-by-step algorithms). In addition, we selected four racing distances commonly contested on outdoor oval tracks (1,500 m, 3,000 m, 5,000 m and 10,000 m).

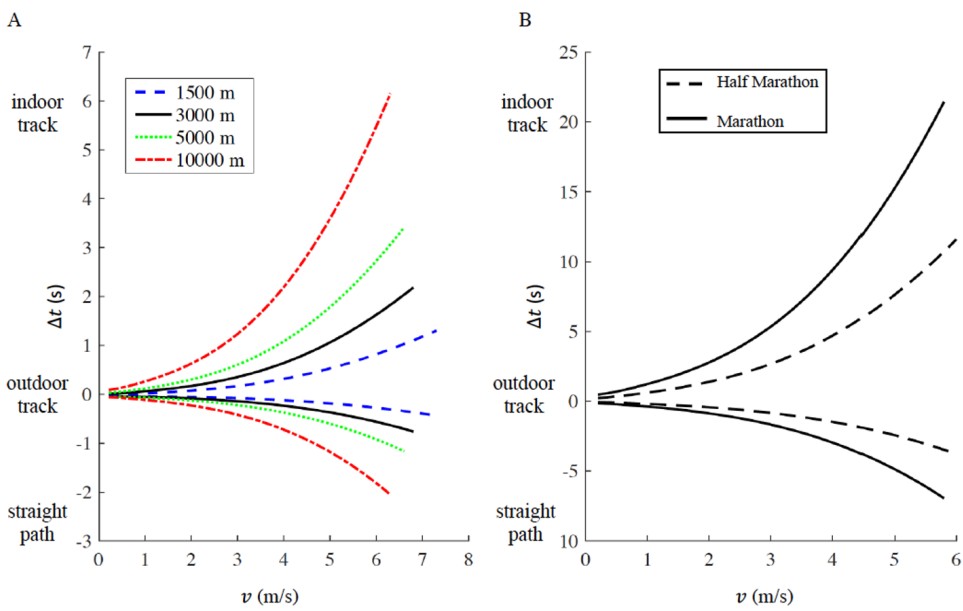

**Figure 4** **Time difference (delta t) for a given racing distance (1,500 m, 5,000 m and 10,000 m in panel A, half marathon and marathon in panel B) as a function of velocity (v).** For a given racing distance, run on a 400 m outdoor track (curve radius $r_{outdoor}$ = 36.8 m), we calculated how much time would increase (delta $t > 0$) on a 200 m indoor track (curve radius $r_{indoor}$ = 17.5 m), or decrease (delta $t < 0$) on a straight path. For each racing distance, the maximum velocity corresponds to the respective current men's world record.

Based on the actual outdoor 400 m track world records for both males and females, we calculated the respective time the same athlete would have run on an indoor track and on a straight racecourse, while keeping the same rate of oxygen uptake maintained on the outdoor track (Table 1). For comparison, we report the actual world record on the indoor track for 1,500 m, 3,000 m and 5,000 m distances. 10,000 m is not officially run on an indoor track (*IAAF, 2018*).

According to our model, an athlete running a marathon in 2:01:39 (corresponding to the actual world record, $v = 5.78$ m/s) on an outdoor track, would run 2:01:32 on a straight path and 2:02:00 on an indoor track. According to our model, an athlete running a half marathon in 58:01 (corresponding to the actual world record, $v = 6.06$ m/s) on an outdoor track, could run 57:57 on a straight path and 58:13 on an indoor track (Fig. 4B).

### Ideal geometry of 400 m track

For a track constrained to comprise a 400 m lap, we report the time difference $\Delta t$ a function of curve radius $r$ in Fig. 5 for 1,500 m, 3,000 m, 5,000 m, and 10,000 m respectively. The plots intersect at ($r = 36.8$ m, $\Delta t = 0$ s), where $t(r) = t_{WR}$. For $r < 36.8$ m, $\Delta t > 0$ s, indicating that a reduction in curve radius, compared to standard outdoor tracks, is detrimental for performance. For example, when $r = 6$ m, $\Delta t$ values range between +1.18 s for 1,500 m and +7.14 s for 10,000 m. On the other hand, for $r > 36.8$ m, $\Delta t < 0$ s for all distances, indicating that an increase in curve radius, compared to standard outdoor tracks, favors

**Table 1** Actual and predicted world records at various racing distances for males (A) and females (B). A 400 m outdoor track is used as the baseline reference for the record predictions on a 200 m indoor track and straight path races. The curve radii are the actual radii run by athletes for indoor ($r = 17.5$ m, (IAAF, 2008a)) and outdoor tracks ($r = 36.8$ m, (IAAF, 2008b)) respectively. 10,000 m is not officially run on an indoor track (IAAF, 2018).

**A**

| Males | 1,500 m | | 3,000 m | | 5,000 m | | 10,000 m | |
|---|---|---|---|---|---|---|---|---|
| Indoor track ($r = 17.5$ m) | Actual record: 3:31.04 | **Predicted record: 3:27.32** | Actual record: 7:24.90 | **Predicted record: 7:22.88** | Actual record: 12:49.60 | **Predicted record: 12:40.78** | Actual record: N.A. | **Predicted record: 26:23.78** |
| Outdoor Track ($r = 36.8$ m) | 3:26.00 | | 7:20.67 | | 12:37.35 | | 26:17.53 | |
| Straight | | **Predicted record: 3:25.59** | | **Predicted record: 7:19.93** | | **Predicted record: 12:36.20** | | **Predicted record: 26:15.47** |

**B**

| Females | 1,500 m | | 3,000 m | | 5,000 m | | 10,000 m | |
|---|---|---|---|---|---|---|---|---|
| Indoor track ($r = 17.5$ m) | Actual record: 3:55.17 | **Predicted record: 3:51.09** | Actual record: 8:16.60 | **Predicted record: 8:07.87** | Actual record: 14:18.86 | **Predicted record: 14:13.77** | Actual record: N.A. | **Predicted record: 29:22.32** |
| Outdoor Track ($r = 36.8$ m) | 3:50.07 | | 8:06.11 | | 14:11.15 | | 29:17.45 | |
| Straight | | **Predicted record: 3:49.76** | | **Predicted record: 8:05.53** | | **Predicted record: 14:10.30** | | **Predicted record: 29:15.88** |

performance; in particular, at the maximum radius ($r = 63.66$ m) $\Delta t$ equals $-0.15$ s for 1,500 m, $-0.31$ s for 3,000 m, $-0.48$ s for 5,000 m and $-0.86$ s for 10,000 m

### Breaking 2 on a straight path

We report the velocities $v_c$ on each of the curve portions and the velocity $v_s$ on the straight portions, calculated assuming that Kipchoge maintained a constant oxygen uptake ($\dot{V}O_{2c} = \dot{V}O_{2s}$) in Table 2. Note that combining each velocity with the respective distance, the time for a full lap (2424.4 m) is $t_{lap,Monza} = 415.1s$, and the total time for a full marathon (17 full laps plus the remaining 0.4 laps, i.e., 980.2 m on the last straight portion) coincides with $t_{Monza} = 7225$ s.

To calculate the time $t_{straight}$ that Kipchoge could have run on a straight marathon course, it is sufficient to divide the total distance by the velocity on the straight:

$$t_{straight} = \frac{d}{v_s} = \frac{42195}{5.8414} = 7223.48s \tag{16}$$

leading to an overall time difference of only $\Delta t = 1.52$ s.

### DISCUSSION

According to our model, the increased time $\Delta t_{180°}$ for an out-and-back race course (i.e.: with a single 180° turn) is less than 0.27 s even in the worst-case scenario (high velocity, 7.3 m/s, and small curve radius, 2.7 m). Nevertheless, race organizers trying to keep $\Delta t_{180°}$ to

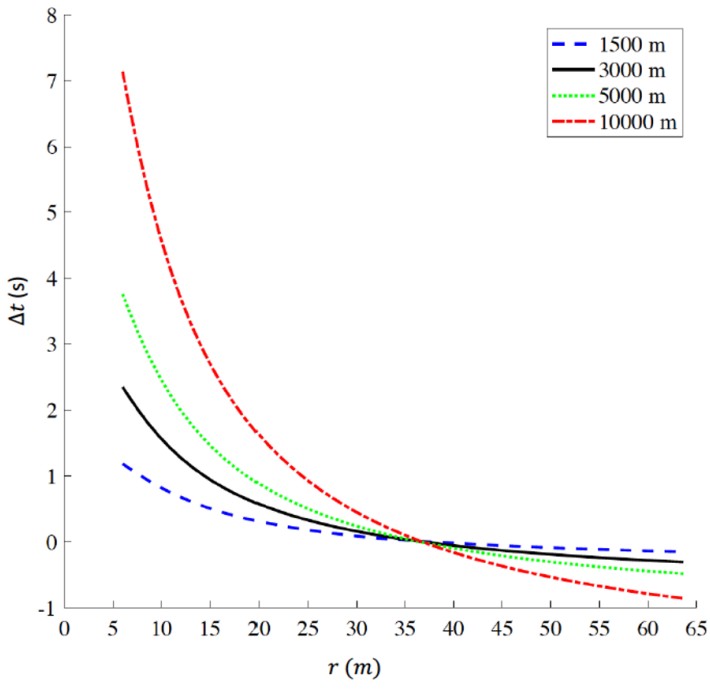

**Figure 5 Time difference (delta $t$) for various racing distances (1,500 m, 3,000 m, 5,000 m, and 10,000 m) as a function of radius (r) for tracks constrained to be 400 m lap distance.** When $r = 36.8$ m, delta $t = 0$ s corresponding to the respective world records on a standard 400 m outdoor track. delta $t > 0$ indicates that any radius $r < 36.8$ m is detrimental for performance, while delta $t < 0$ indicates that any radius $r > 36.8$ m favors performance. The ideal geometry for a 400 m track is a perfect circle with radius $r = 63.66$ m. (Note: $r$ is the actual radius that includes the 0.3 m offset from the inner edge of the curb, to take in account the theoretical line of running according to IAAF rules (*IAAF, 2008b*).

**Table 2 Radii and distances of each of the six curves we identified for the Monza racetrack.** The Straight row represents the sum of all the straight portions of the racetrack. For each portion, we calculated the running velocities on the straight and on each of the curved portions of the track assuming that Eliud Kipchoge maintained a constant $VO_{2c} = VO_{2s}$ (see algorithm 2 in Appendix for details).

| Curve # | Radius (m) | Distance (m) | Velocity (m/s) |
|---|---|---|---|
| 1 | 23 | 32 | 5.8165 |
| 2 | 24 | 21 | 5.8185 |
| 3 | 25 | 62 | 5.8202 |
| 4 | 350 | 116.7 | 5.8413 |
| 5 | 164 | 151 | 5.8409 |
| 6 | 80 | 134.7 | 5.8393 |
| Straight | | 1,907 | 5.8414 |

a minimum, should aim for the largest curve radius allowed by road widths, the presence of buildings/sidewalks, median strip etc.

IAAF rules 2018 require that, in order for a race course to be record-eligible, the start and finish points of any road race shall not be further apart than 50% of the total race distance (Rule 260.21), making the presence of at least one curve mandatory for all record-eligible

courses. While having the largest possible radius is still a valid recommendation, in races measuring 5,000 m and longer, athletes can maintain a lower velocity compared to the scenario described above (6.6 m/s being the average velocity for Bekele's 5,000 m world record), and would therefore experience even lower values for $\Delta t_{180°}$. In addition, $\Delta t_{180°}$ becomes trivial in terms of percentage of the total race time especially in races like the half marathon or the marathon, while other factors, like change in elevation (*Giovanelli et al., 2016*; *Hoogkamer, Taboga & Kram, 2014*), surface type (*Kerdok et al., 2002*), drafting (*Hoogkamer, Snyder & Arellano, 2019*) etc., have a much greater effect on running energetics and therefore on the overall time (*Hoogkamer et al., 2016*). For example, *Hoogkamer, Kram & Arellano (2017)* estimated that the maximum allowable downhill elevation drop (42 m) in a marathon could save 28 s, and a legal tailwind could save ∼3 min. More economical running shoes should save roughly 3 min (*Kipp, Kram & Hoogkamer, 2019*).

Lacking empirical data from a controlled study, we can only evaluate the validity of our model by comparing our predictions to actual race performances. Starting with the outdoor 400 m track records, our model predicts faster indoor world records compared to the actual record times in all distances for both males (Table 1A) and females (Table 1B).

The differences between our model predictions and the actual indoor world records, based on the times run on 400 m outdoor tracks, are 3.75 s (1.78%) in the 1,500, 2.2 s (0.45%) in 3,000 m, and 8.82 s (1.14%) 5,000 m for males (Table 1A), and 4.1 s (1.74%) in the 1,500, to 8.73 s (1.76%) in the 3,000 m and 5.09 (0.59%) in the 5,000 m for females (Table 1B). It must be noted that multiple factors can contribute to these differences between predicted and actual times. Indoor races are typically run in winter, while outdoor races are run in spring/summer and athletes tend to reach peak fitness for outdoor races when major international competitions (Olympics, World Championship etc.) are held. Only the men's 5,000 m indoor and outdoor records were run by the same athlete (Kenenisa Bekele) in the same year. All other indoor and outdoor records were run by the same athlete, but in different years, or by different athletes. Pacing and drafting play important roles when trying to run a world record time (*Hoogkamer, Snyder & Arellano, 2019*); it is likely more difficult for athletes on indoor tracks to negotiate the smaller curve radiuses while following or overtaking other competitors, compared to outdoor tracks.

Our model, and Table 1, can be used to identify which, among the actual indoor world records, is the hardest or easiest to break, assuming the outdoor world record is a "benchmark performance" corresponding to the current "physiological limits" of males and female athletes respectively. Our model, in fact, calculates what time an athlete with the exact same fitness level and all the conditions (drafting, motivation etc.) found during the outdoor world record could run on tracks with different curve radiuses or on the straight. For males, it is evident that Daniel Komen's 3,000 m indoor world record is only 0.45% slower compared to the "physiological limit" he himself reached on the outdoor track two years earlier. In order for an athlete to break the indoor world record, they must be close to being able to run under the current outdoor world record. On the other hand, the current 1,500 m indoor world record is 1.78% slower compared to the "physiological limit" set by Hicham el Guerrouj on an outdoor track and seems therefore relatively easier to break. In order to break the current indoor world record by 0.01 s, an athlete must be able to run

3:29.80 on a 400 m outdoor track. For females, the 5,000 m indoor world record is only 0.59% slower compared to the "physiological limit" set by Tirunesh Dibaba on an outdoor track. The women's outdoor 5,000 m record is relatively harder to break compared to the 3,000 m indoor world record, 1.76% slower compared to the "physiological limit" set by Junxia Wang on a 400 m outdoor track.

Our prediction that a perfectly circular track is optimal for distance running performance concurs with Greene's model for sprint running (*Greene & Monheit, 1990*). This is true for all racing distances. A 1,500 m runner is more affected by the velocity reduction on the curve ($v_c << v_s$) because of their faster average velocity compared to longer distances. However, runners competing in longer events have to perform more laps around the track (up to 25 laps for the 10,000 m). The number of laps seems therefore the dominant factor on the overall increased/decreased time as a function of curve radius. The time difference between a standard outdoor track ($r = 36.8$ m) and a perfectly circular 400m track ($r = 63.66$ m) according to our model ranges between $-0.15$ s for 1,500 m, and $-0.86$ for 10,000 m. When designing a stadium, an efficient use of the available space is critical: a standard track allows for a rectangular field in the center that can be used for multiple sports (football, soccer etc.) with a trivial sacrifice in terms of running performance. Interestingly, Australian football is played on an oval field that could accommodate a perfectly circular track on the outside and provide a direct confirmation of our predictions. It would also be interesting to see if and how a circular track, compared to the standard track, could influence race tactics.

Some insight into the validity of our model can also be gained by comparing the best performances of two world-class athletes (Jenny Simpson and Sydney Maree) when racing one mile (1,609 m) indoors, outdoors and on a straight racecourse (5th Avenue Mile, NY). Considering their outdoor personal best as their "benchmark performance" (4:17.30 for Simpson, 3:48.83 for Maree), our model predicts times on a 200 m indoor track of 4:18.27, 2.87% faster than the time of 4:25.91 run by Simpson, and 3:50.10, 0.99% faster compared to the time of 3:52.40 run by Maree. For a straight race, our model predicts times of 4:16.98, only 0.14% slower compared to the actual time of 4:16.6 run by Simpson and 3:48.40, only 0.39% slower compared to 3:47.52 run by Maree. While the same considerations highlighted above must be taken in account when comparing different races (different years or racing seasons, different fitness levels), we must also take in account that the 5th Avenue Mile is slightly net downhill which may explain why both our predictions seem slower than the actual race times.

The NCAA indoor track time conversion system provides another validity test. The NCAA conversion factors were developed using thousands of race performances, comparing times of the same athlete in different indoor facilities (*Pederson et al., 2012*). However, these conversions do not specifically take in account the exact curve radius of each indoor track. Rather, the NCAA categorizes them as "undersized" (<200 m per lap, like the Madison Square Garden track, which is 146.3 m per lap (*Attwood, 2012*), "standard" (200 m per lap) and "oversized" (>200 m per lap, typically 300 m (*Pederson et al., 2012*). In addition, racing velocity is accounted for only in terms of male vs. female athletes and in terms of racing distances. For example, as the racing distance increases from 200 m to 5,000 m, the NCAA

**Table 3  Comparison of the NCAA conversion factors vs. the current proposed model for 3,000 m and 5,000 m racing distances for males (A) and females (B).** For each distance and for each sex, we identified sample performances on a standard indoor track (200 m per lap, curve radius $r = 17.5$ m) and converted them to an undersized track (corresponding to the Madison Square Garden indoor track, 146.3 m per lap and $r = 11.7$ m (*Attwood, 2012*)) and to an oversized track (300 m per lap, $r = 35$ m, https://www.pl-linemarking.co.uk/300-metre-track-line-marking.html) using NCAA conversion tables and our model. Performances are reported in minutes:seconds.hundredths.

**A**

| Males | 3,000 m | | 5,000 m | |
|---|---|---|---|---|
| Undersized indoor track (1 lap 146.3 m, $r = 11.7$ m) | NCAA: 8:02.27 | Current model: 8:02.28 | NCAA: 14:03.29 | Current model: 14:03.45 |
| Standard indoor track (1 lap = 200 m, $r = 17.5$ m) | 8:00.00 | | 14:00.00 | |
| Oversized indoor track (1 lap = 300 m, $r = 35$ m) | NCAA: 7:54.50 | Current model: 7:58.40 | NCAA: 13:51.11 | Current model: 13:57.63 |

**B**

| Females | 3,000 m | | 5,000 m | |
|---|---|---|---|---|
| Undersized indoor track (1 lap 146.3 m, $r = 11.7$ m) | NCAA: 9:01.03 | Current model: 9:01.79 | NCAA: 16:01.06 | Current model: 16:02.60 |
| Standard indoor track (1 lap = 200 m, $r = 17.5$ m) | 9:00.00 | | 16:00.00 | |
| Oversized indoor track (1 lap = 300 m, $r = 35$ m) | NCAA: 8:55.40 | Current model: 8:58.79 | NCAA: 15:52.66 | Current model: 15:58.25 |

conversion factor from "oversized" to "standard" indoor tracks decreases from 1.0179 to 1.0107 for males, and from 1.0155 to 1.0077 for females. Despite these limitations, we can compare the NCAA conversions with our model predictions for 3,000 m and 5,000 m for male (Table 3A) and female (Table 3B) athletes.

Comparing standard 200 m vs. undersized indoor tracks, the difference between our model and NCAA conversions range from 0.01 s (<0.01%) for males in the 3,000 m, to 1.54 s (0.2%) for females in the 5,000 m. Comparing standard vs. oversized indoor tracks, the differences between the current model and NCAA conversions range from 3.39 s (0.63%) for females in the 3,000 m, to 6.52 s (0.78%) for males in the 5,000 m. While both NCAA conversion tables and our models agree on the overall effect of smaller vs larger radii on performance (i.e., the larger the radius, the better the overall time), our model predicts a slightly greater time when going from standard to undersized tracks, while it predicts smaller time reductions when going from standard to oversized tracks compared to NCAA conversion tables. These differences could be explained by the fact that NCAA tables provide an average conversion factor for a given race, independent of the actual performance of the athlete in that race, while in our model, velocity has a non-linear effect on the decreased or increased time on tracks of different radii (see Fig. 4).

When Eliud Kipchoge participated in the Breaking 2 attempt, he completed 17.4 laps around the Monza racetrack, totaling 105 curves (note that we divided the "Curva parabolica" into three sections, but even considering it as one single curve the total number of curves would still be 71). Our model predicts a trivial 1.52 s time difference between the Breaking 2 attempt and a marathon run on a straight racecourse. This is due to the fact that the smallest radius on the Monza racetrack is still 23 m (curve #

1), a value 31% bigger than the radius of indoor tracks (17.5 m), therefore the velocity reduction on curves is hardly noticeable. A similar number of curves can be counted for two of the most famous marathon racecourses: ∼50 curves for the Berlin marathon (https://www.bmw-berlin-marathon.com/en/your-race/start-course-finish/course/) and ∼70 for the London marathon (https://www.virginmoneylondonmarathon.com/en-gb/event-info/runner-info/). Even though we could not measure the radii of these curves, our model predicts that the increased time due to curve negotiation, compared to a straight racecourse, is negligible and the general perception of the magnitude of the effects of curves on road racing performance is not supported by our calculations.

## Limitations and future studies

Running economy is affected by a multitude of biomechanical factors. In combination with the axial leg force that drives our model, contact time of the foot with the ground and the rate of force production (*Roberts et al., 1998*), antero-posterior ground reaction forces (*Chang & Kram, 1999*), stride length (*Cavanagh & Kram, 1989*) and stride frequency (*Snyder & Farley, 2011*) all affect the energetic cost of running. When running at maximum speed on curves with small radii ($r \leq 6$ m), runners increase their contact time, decrease antero-posterior ground reaction forces and stride length compared to straight running (*Chang & Kram, 2007*). It is unclear if these biomechanical differences are maintained at sub-maximal speeds and at the larger radii. We have no knowledge of studies that measured biomechanics and/or, more crucially, energetics of curve running that could validate our model. In the future, we intend to empirically test a key assumption of our model - that athletes run slower on curves compared to straight portions of a track during races. Indeed, it is not clear if athletes can accurately sense their speed and metabolic rate with the precision and time resolution required. It may be that athletes run at the same speed on straight and curved sections and thus do not maintain a constant metabolic rate.

The data collected by *Teunissen, Grabowski & Kram (2007)* that allowed us to derive Eq. (1) were collected at one fairly slow velocity (3 m/s) on a treadmill, but to our knowledge there are no equivalent data for faster running velocities and none for overground running under different gravity conditions. In addition, we extrapolated Eq. (1) beyond normal gravity, assuming the same slope is maintained when the average axial force acting on the runner is increased ($\overline{F}_a > 1BW$). Additional experiments are needed to quantify the effects of different velocities and increased gravity on Eq. (1) and verify our assumption. In addition, our model does not distinguish between male and female athletes. While Eq. (1) can be applied to both male and female athletes, given that *Teunissen, Grabowski & Kram (2007)* included both sexes in their study, Eq. (7) was derived for male runners only (*Kipp, Kram & Hoogkamer, 2019*). Generally, studies find that males are slightly more economical than females at matched absolute running velocities (*Daniels & Daniels, 1992*). Equation (7) should therefore be adapted for female athletes with a different set of parameters that take in account these differences.

Our extrapolation of the *Kipp, Kram & Hoogkamer (2019)* equation seems reasonable but it would be preferable to obtain empirical measurements of oxygen uptake or metabolic rate for elite athletes at faster running speeds that are closer to the world

record performances. In order to run 42,195 m on the Monza racetrack in a total time $t_{Monza} = 7225$ s or 5.84 m/s, Eq. (7) predicts that Eliud Kipchoge sustained a rate of oxygen uptake $\dot{V}O_{2s} = 80.87$ mlO$_2$/min/kg (see algorithm 2 in Appendix for details). This incredible value suggests that either that the *Pugh (1970)* factor for air resistance is too large or that Kipchoge is much more economical runner that the subjects tested by *Kipp, Kram & Hoogkamer (2019)*. Fortunately, the absolute value does not affect our calculations of the effects of curve running.

When we model an athlete transitioning from straight to curved running, such as when running on a track or on a non-straight road race, we assume that the change in velocity (from $v_s$ to $v_c$ and vice-versa) is instantaneous, i.e., there is no deceleration or acceleration phase between straight and curved portions. This assumption may be reasonable for larger radii, such as outdoor or indoor racing tracks. If $r = 17.5$ m, when the velocity on the straight is $v_s = 7.00$ m/s, the velocity on the curve is reduced to $v_c = 6.91$ m/s, allowing an athlete to decelerate and re-accelerate in one single step. But, for much smaller radii (e.g., $r = 1$ m) when the velocity on the straight is $v_s = 7.00$ m/s, the velocity on the curve is $v_c = 4.69$ m/s, an athlete would likely need more than one step to decelerate and then re-accelerate). Non-trivial decelerations and accelerations increase the metabolic cost of running (*Di Prampero et al., 2005*) and should therefore be factored into our model, especially for smaller (<6 m) curve radiuses. This approach, while theoretically possible, can lead to accurate calculations only if the exact values of deceleration and accelerations are known. Future studies (from video and/or from lab-based measurements) could provide such information and fill this gap to create a more realistic model.

When an athlete is running on a curve with large radius, even for faster (>7 m/s) velocities, the increase in axial force $\overline{F}_a$ is relatively small (see Fig. 1B) and it is reasonable to assume that, as modeled in this paper, the limiting factor on $v_c$ is mainly the metabolic cost of running. However, at smaller radii the increase in $\overline{F}_a$ is much more marked (for $r = 1$ m, when $v_s = 7.00$ m/s and $v_c = 4.69$ m/s, $\overline{F}_a = 2.45$ BW). $\overline{F}_a$ is the axial force calculated over a full step, assuming a duty factor of 45% (*Chang & Kram, 2007*) the average force during contact reaches an even higher value of 2.72 BW. *Chang & Kram (2007)* measured velocities and ground reaction forces of recreational athletes sprinting on the straight and on curves of small (6 m or less) radii. While subjects were able to reach $v_s = 7.70$ m/s on the straight, the maximum velocity on a curve when $r = 1$ m was only 2.99 m/s, well below the velocity predicted by our model. In addition, the *peak* axial forces reached only 1.87 BW for the inside leg and 2.25 BW for the outside leg. In the *Chang & Kram (2007)* study, subjects were instructed to run as fast as possible but only for a very limited amount of time. Therefore, the maximum velocity they were able to attain on curves was not limited by metabolic cost, but by other constraints. *Chang & Kram (2007)* concluded that during small radius curve sprinting, the ability to generate force, in particular from the inside leg, limits maximum curve velocity. When athletes run on curves at sub-maximal speeds (i.e., for a prolonged period of time), it is likely that both mechanisms play roles. When transitioning from straight to curved running, at larger radii, the main driving factor in the velocity reduction is maintaining a constant metabolic rate. But, at progressively smaller

radii, the increase of centripetal, and therefore axial forces, is amplified and velocity is further reduced as the athlete is limited by his/her ability to generate forces.

A third effect not included in our model is the difference between flat and banked curves. Equations (1) and (7), in particular, apply only to straight running. When we calculate the average axial leg force $\overline{F}_a$ of an athlete running on a curve, we assume that the body is aligned with $\overline{F}_a$ (see Fig. 1, inset) and the ankle remains in the sagittal plane, similar to straightline running. In other words, we assume that the curve is banked and $\overline{F}_a$ is always perpendicular to the running surface. Greene (1987) provided evidence that this is the ideal condition for sprint running and showed that any deviation of the banking angle, resulting in a misalignment of $\overline{F}_a$ and the running surface, results in a further reduction of velocity on the curve $v_c$. For example, if $v_c$=10 m/s and $r = 17.5$ m, the angle of $\overline{F}_a$ with the vertical is 30°. A sprinter running on a flat (unbanked) curve would slow down by 1.1 m/s compared to a curve with a 30° bank (Greene, 1987). Since this effect is much less important at distance running velocities (<7 m/s) and for large radii (e.g., Monza racetrack), we did not account for this further reduction in $v_c$ on flat, or non ideally-banked, curves.

## CONCLUSIONS

Our model assumes that runners reduce their velocity on curves, compared to straight running, to maintain a constant metabolic rate for the whole duration of the event. This reduction is marked for smaller curve radii, such as indoor tracks, and at faster velocities. The effect becomes negligible, in terms of overall performance, for larger radii and slower speeds, such as those seen in city marathons. The general perception of the magnitude of the effects of curves on road racing performance is not supported by our calculations

## ACKNOWLEDGEMENTS

The authors thank Brett Kirby, Ph.D. for providing the blueprints of Monza racetrack, and Owen Beck, Ph.D. and Shalaya Kipp, M.S. for their helpful critiques of an earlier version of this manuscript.

### Funding

The University of Colorado Boulder Libraries funded the open access fee through the institutional arrangement with PeerJ. The funders had no role in study design, data collection and analysis, decision to publish, or preparation of the manuscript.

### Grant Disclosures

The following grant information was disclosed by the authors:
The University of Colorado Boulder Libraries.

### Competing Interests

The authors declare there are no competing interests.

## Author Contributions

- Paolo Taboga performed the calculations, analyzed the data, contributed materials/analysis tools, prepared figures and/or tables, authored or reviewed drafts of the paper, approved the final draft.
- Rodger Kram conceived as the approach, analyzed the data, authored or reviewed drafts of the paper, approved the final draft.

## Data Availability

Matlab codes to calculate velocity on the curves and to convert from outdoor track to straight races and indoor tracks are available in the Supplemental Files.

## Supplemental Information

Supplemental information for this article can be found online at http://dx.doi.org/10.7717/peerj.8222#supplemental-information.

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
