# Peer review of "Modelling the effect of curves on distance running performance"

_PeerJ, doi:10.7717/peerj.8222_

## Round 0.1 · original submission · Minor Revisions

Apologies for some delay in obtaining reviews. However, the good news is that all 3 reviewers find value in the study and feel the science is strong enough. They have constructive critiques on numerous points; each one rather different for each reviewer; that will require moderate revisions. Re-review might not be necessary if you detail and justify your amendments clearly. Thank you for submitting this interesting paper-- we look forward to the revised MS.

Reviewer 1 ·

Basic reporting

Clear, mostly unambiguous (see comments later), good lit review though I suggest adding one more. Good structure, but Figs 6 and 7 not cited in the main text.

Experimental design

Modelled. Well explained in the main. The work is presented as descriptive rather than hypotheses-driven.

Validity of the findings

Insightful. Could be 'sold' better.

Additional comments

I enjoyed reading this manuscript, which does a pleasingly good job of walking the typical non-mathematically-minded biologist (such as myself) through the equations. I have few things to fault in the report, although I feel that the Abstract and opening to the Discussion are a little prosaic – I would encourage the authors to consider ‘selling’ their manuscript a little better. For example, perhaps stating that their model indicates that indoor track records are underperformances compared to outdoor records, even when accounting for the tightness of the bends. And that the 2-hr marathon record would not have been beaten even if the route had been entirely straight. That marathon runners lose very little time on most courses due to their curves. That the standard outdoor track dimension is arguably a close to optimal trade-off between curve-effect-minimisation and space efficiency. Things like that.

The following is a relevant paper that probably should be cited in this manuscript – it shows that for the same pace of movement in humans (albeit walking), metabolic rate is higher when turns are included, particularly when those turns are more acute:
Wilson, R., Griffiths, I., Legg, P., Friswell, M., Bidder, O. R., Halsey, L. G., . . . Shepard, E. (2013). Turn costs change the value of animal search paths. Ecology Letters, 16(9), 1145-1150. doi:10.1111/ele.12149

L50: Here or elsewhere, I think it worth noting along which dimension that curve radius is measured. For some reason, in my head I was initially seeing it as a measure of how far beyond the ends of the straights the curve extended, rather than the ‘width’ of the curve i.e. how far apart the straights are. I believe the latter is what is really meant. Of course, a ‘wide radius’ means the opposite for these two concepts of the curve radius. Indeed, perhaps as an extra panel to one of the Figures, it might be useful to show a schematic of a few different track dimensions depending upon the curve radii, to help the reader (particularly those not accustomed with track and field) visualise the concept being presented in this manuscript. … Also, is it the case/assumed that the curves are always the same length regardless of their curvature?

L53: ‘for’ missing

L174: It could be worth quantifying this. E.g. how much difference is there assuming 100% glucose utilisation versus perhaps 50% glucose, 25% fat and 25% protein (i.e., I believe two feasible scenarios)?

L180: ‘this’ missing.

L184: Is this assuming V’O2c = V’O2s?

L186: be clear here that the ‘180 degree turn’ is clinal rather than instantaneous. In my mind I saw a person stopping, turning 180 degrees and then running again.

L187: period punctuation mark required here?

L208: presumably assuming consistent V’O2

L213: aren’t indoor tracks often banked? How might this affect calculations? Maybe something for the Discussion.

L216: calculate

L229: consumption better than uptake?

L241: semi colon better than a comma here, given how the sentence is constructed.

L308: This surprises me – the marathon runner only loses 7 seconds across the entire race because of the (many) bends he encounters on the track. This suggests that the 3 second difference in NCAA mile time conversion is spurious, most of the loss of speed on the indoor track being due to weaker physical performances rather than the additional and sharper bends (though I appreciate the miles runners are moving faster than marathon runners, accentuating the ‘cost’ of the bends).

L313: While a radius greater than 36 m would appear to be advantageous, it seems to be only slight. Might this suggest that the standardised 36 m might not be coincidence? I.e. there was some trade-off made, albeit perhaps not quantified, between limiting the effects of curve running versus space efficiency?

Figure 1 caption: panels C and D not cited.

Figure 1: (panel B) when running a curve of 36.8 m radius, at any speed the additional axial force involved is always just fractionally above 0 (i.e. Fa = 1.1BW)?

Figures 6 and 7 are not cited in the main text!

·

Basic reporting

no comment

Experimental design

no comment

Validity of the findings

no comment

Additional comments

Summary: This manuscript describes a modeling study to predict the effect of curves on the performance of runners as it pertains to running velocity and total race time. It uses the rationale that runners will maintain a constant metabolic rate, which will force them to slow down on curves as a function of the radius of curvature and axial leg force requirements. Overall, this is a very well written manuscript that is methodical in its logic and execution. The authors should be commended on a fine study. I do have some suggestions that may help with discussion points, however, my overall impression of the manuscript is very positive.

1) The first model assumption, which the authors point out as a limitation in lines 446-457, is that runners decrease velocity on curves during race performance. Decreased velocity on the curve for races of these distances at submaximum speed is unknown, as the authors point out. However, given that their model is trying to predict competitive race performances, it seems quite plausible that racers might choose not to slow down on a curve. Instead, they could suffer some added metabolic cost in order to maintain their speed relative to their competition. One aspect of this, which was not discussed is whether and how a runner could even control and maintain a constant metabolic rate. Although a person might be able to sense overall metabolic energy expenditure over long distance and time (on the order of minutes), they would need to negotiate a curve in a much faster timeframe than their metabolic rate could be sensed (on the order of seconds). In contrast, something like running velocity is rather easy for a runner to sense and control. So one might expect a runner could maintain their velocity in a race at the expense of a transient increase in energy cost. As the authors mentioned, what runners actually do is unknown and would benefit from a study on the subject, but it should be worth noting that sensing and controlling metabolic rate over something like running velocity may be a difficult task.

2) Related to #1 above, the second assumption, which the authors also address in lines 459-466 as a limitation, is that these runners maintain a constant metabolic rate througout the race and that non-oxidative sources of energy are not accounted for in their model. Although runners may be running at submaximum velocity, is it correct to assume they are running with negligible non-oxidative metabolic costs even for longer distance races? As competitive races go, how robust is the assumption that runners would be pushing their physiological limits while not taking advantage of additional non-oxidative sources of energy? Even if it is to maintain speed on a curve, or perhaps to maintain or increase running velocity at the end of a race, it would seem that this assumption might warrant more discussion on how non-oxidative metabolism might change the predictions.

3) Lines 459-461: Related to above, #2. What is the reference or rationale for the statement that nearly 100% of energy for longer races comes from oxidative sources? I believe the Hill 1999 paper only addressed up to a 1500m distance where non-oxidative sources could be as high as 20%.

4) There is another important assumption about linearity of the increase in metabolic cost as average axial leg force increases. Again, the authors did a good job of citing this as a limitation in their model, but can they (generally) comment on how their predictions might change if this relationship were non-linear? For example, if metabolic cost increased greater than linearly with increasing leg force, how would this affect your model predictions? The rationale for this comment is that as one recruits faster, less economical muscle fibers at higher forces, the metabolic rate per unit of force generated may increase at a faster rate than at lower forces.

5) Line 55, just to improve clarity, please replace “…total axial leg force…” with “…average axial leg force…”

6) Lines 102-110: I understand what the point being made here is, but as pointed out later in the manuscript, there are non-frontal plane forces that runners are also subject to (i.e., braking and propulsive forces). Please revise to clarify that this section is addressing forces only in the frontal plane that runners are subject to.

7) Equations 7 and 8: This is a very minor point, but the constant in both equations should be identical, but they have a different number of significant digits (18.91 vs 18.911).

Reviewer 3 ·

Basic reporting

Line 545 - ‘At faster velocities, … slower speeds ...’
The wording of this sentence is contradictory, and should be revised.

Perhaps a surface/contour plot of figure 1, with some of the corners/speeds considered in the discussion indicated on the plot would be more illustrative, plotted over a more relevant range.

The model assumes perfect traction at the runner's feet. At what point could the traction be limiting to cornering ability, and contribute to a speed decrease? The same question applies to ankle strain, at what point is fear of injury limiting?

Does traction limit speed when going around the bend? The model does not appear to account for potential changes to traction (according to the model different conditions of traction would not affect running speed).

The manuscript would benefit from conveying better how the findings are relevant to running in other model systems beyond a narrow human biomechanics viewpoint. Can authos better cover a comparative context in their introduction and their discussion.

When it comes to running around a bend a controversial question in the field is what factors limit performance.

In the interest of discussing a comparative aspect better, authors could discuss other studies comparing performance of running on a straight line versus curve running.

What effect would authors anticipate from comparison of performance under locomotion around the bend with a camber (incline) in the curved section of the trackway?

In contrast to findings from the paper under consideration, Usherwood and Wilson 2005 (cited by authors) suggested that greyhounds would not suffer any reduction of top speed when going around a bend with a camber.

By contrast, experiments by Hayati et al (2017) based on IMU and foot print analyses found that greyhounds do slow down when going around the bend. Since the latter experiment is more line with performance described in the manuscript under consideration, it would be imperative that authors discussed its findings in comparison with their own.

In this spirit, the discrepancy between these studies requires further consideration and discussion sensu Hayati H., Eager D., Jusufi A., Brown T. (2017) A Study of Rapid Tetrapod Running and Turning Dynamics Utilizing Inertial Measurement Units in Greyhound Sprinting. AMSE DETC 67691, V003T13A006.

Experiments on human locomotion around the curve while suspended suggest that traction management is a key feature to the speed reduction.

Experimental design

Could the authors give some further indication of the validity of the empirical equations 1 and 7, as they are critical to the rest of the analysis?

Validity of the findings

The idea and execution is not unsound, and the paper is well written. I find the evidence is not sufficient to validate the model and its underlying assumptions, or at least to demonstrate its utility. But the authors are candid about the limitations of their analysis, and as a first step to further study it is a reasonable, if incremental contribution.

Additional comments

The authors describe a model of the cornering energy cost in human running based on combining existing empirical relationships with the basic mechanics of running on a curved path. They use this to estimate speed decreases based on a constant energy consumption. They then compare the predictions this model makes for speed decreases resulting from course curves with observed differences between curved and straight performances by real athletes.

The authors often refer to other factors which affect race times (the presence of a slope, current fitness etc.) but do not give any sense of the magnitude of the effects. It would be helpful to give some measures of spread for competing effects, to have a sense of the relative size of predicted time changes.

Other than the NCAA comparisons, the authors rely on individual events such as records, and I feel the paper would benefit from a more statistical analysis with attention paid to spread / variation.

Why do the authors often choose to compare maximal/record times, whereas the model is based on data from less high performing athletes, and what effect is this expected to have?

---

## Round 0.2 · accepted · Accept

I have read the rebuttal and checked the MS vs. prior version and feel that the authors have done a very good job of responding to reviews and incorporating their constructive critiques into the MS where feasible. I do not judge another round of review to now be necessary. I urge the authors to publish their peer review history with the MS as there are some details that might help readers understand the nuances and perspectives (but are not needed in the MS itself). Congratulations on a fine study.